# Bicuspid Aortic Valve: Old and Novel Gene Contribution to Disease Onset and Complications

**DOI:** 10.3390/diagnostics16010104

**Published:** 2025-12-28

**Authors:** Elena Sticchi, Rosina De Cario, Samuele Suraci, Ada Kura, Martina Berteotti, Lapo Squillantini, Giulia Barbieri, Rebecca Orsi, Maria Pia Fugazzaro, Stefania Colonna, Francesca Gensini, Erika Fiorentini, Anna Maria Gori, Rossella Marcucci, Guglielmina Pepe, Stefano Nistri, Betti Giusti

**Affiliations:** 1Department of Experimental and Clinical Medicine, University of Florence, Largo Brambilla 3, 50134 Florence, Italy; decariorosy@hotmail.it (R.D.C.); samuele.suraci@unifi.it (S.S.); ada.kura@unifi.it (A.K.); martina.berteotti@unifi.it (M.B.); l.squillantini@student.unisi.it (L.S.); giulia.barbieri@unifi.it (G.B.); r.orsi2@student.unisi.it (R.O.); annamaria.gori@unifi.it (A.M.G.); rossella.marcucci@unifi.it (R.M.); guglielminapepe@hotmail.it (G.P.); betti.giusti@unifi.it (B.G.); 2Marfan Syndrome and Related Disorders Regional (Tuscany) Referral Center, Careggi Hospital, 50139 Florence, Italy; 3Atherothrombotic Diseases Center, Cardiothoracovascular Department, Careggi University Hospital, 50134 Florence, Italy; 4Cardiology Service, CMSR Veneto Medica, 36077 Altavilla Vicentina, Italy; mariapiafugazzaro@hotmail.it (M.P.F.); stefanonistri41@gmail.com (S.N.); 5Outpatient Cardiology Unit, Health District 1 ULSS 6, Vigonza and Carmignano di Brenta, 35128 Padua, Italy; stefaniacolonna@hotmail.com; 6Medical Genetics Unit, Department of Experimental and Clinical Biomedical Sciences ‘Mario Serio’, University of Florence, 50139 Florence, Italy; francesca.gensini@unifi.it (F.G.); erika.fiorentini@unifi.it (E.F.)

**Keywords:** heart valve diseases, aortic aneurysm, aortic valve stenosis, genetics

## Abstract

**Background:** Bicuspid aortic valve (BAV) is the most common congenital heart defect, and its complications (namely, dilatation of the thoracic ascending aorta) raise concerns regarding the proper timing of aortic surgery. The study aim is to unravel the genetic basis of BAV and its complications through a high-throughput sequencing (HTS) approach and segregation analysis if family members were available. **Methods:** Fifty-two Italian BAV patients were analyzed by HTS using the Illumina MiSeq platform. Targeted sequencing of 97 genes known to be or plausibly associated with connective tissue disorders or aorthopathy was performed. Thirty-five first-degree relatives of N = 10 probands underwent mutational screening for variants identified in the index cases. **Results:** HTS identified 194 rare (MAF < 0.01) variants in 63 genes. Regarding previously reported genes, five *NOTCH1* variants in four BAV patients, four *FBN1* variants in two patients and one *GATA5* variant in one patient were identified. Interestingly, among further loci, the possible contribution of *PDIA2*, *LRP1* and *CAPN2* was suggested by (a) the increased prevalence of rare genetic variants, independently from their ACMG classification in the whole BAV cohort, and (b) segregation analyses of variants identified in family members. Moreover, the present data also suggest the possible contribution of rare variants to BAV complications, specifically *MYLK* in aortic dilatation, *CAPN2* in BAV calcification and *VHL* and *AGGF1* in valve stenosis. **Conclusions:** Our results underline clinical and genetic diagnosis complexity in traits considered monogenic, such as BAV, but characterized by variability in disease phenotypic expression (incomplete penetrance), as well as the contribution of different major and modifier genes to the development of complications.

## 1. Introduction

Bicuspid aortic valve (BAV) represents the most common congenital cardiac defect (0.5–2%) and is the result of the fusion between adjacent aortic cusps during valvulogenesis [1]. BAV genetic determinants have been investigated by a large number of studies in both isolated as well as syndromic presentations. Indeed, a higher BAV prevalence was observed in patients affected by several genetic syndromes, including connective tissue disorders as Marfan, Loeys–Dietz and vascular Ehlers–Danlos syndromes [2,3,4]. From early analyses of aortic valves in genetically engineered mice to more recent next-generation sequencing (NGS) approaches on selected genes, several studies have been carried out to unravel the molecular mechanisms associated with BAV development and complications [5,6,7]. Moreover, the coexistence of BAV with syndromic disorders, increasing both morbidity and mortality [8], strengthens the genetic heterogeneity hypothesis in BAV pathogenesis and severity according to the different associated genes. Indeed, association with the abovementioned syndromes raises the issue of the role of their causative genes (e.g., *FBN1*, 15q21.1; *TGFBR2*, 3p24.1; *TGFBR1*, 9q22.33; *ACTA2*, 10q23.3) in BAV onset and complications. BAV often occurs in association with thoracic aortic aneurysms (TAAs), independently of age and severity of aortic valve hemodynamic impairment. Moreover, an increased incidence of thoracic aortic dissections is reported in BAV [8]. Other clinical consequences associated with BAV might be represented by aortic stenosis and insufficiency [9], as well as calcification [10].

These observations led to the hypothesis of a genetic architecture of BAV in which many different genetic variants interact in an integrated/synergic manner to influence BAV onset and complications [1]. Increasing knowledge on BAV genetic bases could improve clarification of its underlying molecular mechanisms, which could be translated into the clinical practice in order to predict its most feared complication, i.e., TAD, to establish individualized genetic risk profiles and to exclude syndromic traits in patients with suggestive manifestations [6,11]. In this respect, high-throughput sequencing (HTS) provides the opportunity to analyze a wide number of genes in order to confirm and identify novel candidate loci in BAV. Accordingly, the aim of this study is to unravel the genetic bases of BAV and its complications (TAA, calcification, aortic insufficiency and stenosis) using a 97-gene panel targeted HTS approach, focusing on 52 Italian index cases with non-syndromic BAV and using segregation analysis if family members were available.

## 2. Methods

### 2.1. Study Population

Fifty-two patients [34 males (65.4%); median age 42.5 years [interquartile (IQR) range 31–51.5 years]] diagnosed with BAV and referred from outpatient cardiologists to the Center for Marfan Syndrome and Related Disorders (Careggi Hospital, Florence) and the Department of Experimental and Clinical Medicine (University of Florence) for clinical evaluation and differential diagnosis were included in the study. A multidisciplinary clinical evaluation is routinely performed for these patients, including cardiovascular investigation/imaging, ophthalmologist investigation, physical examination and genetic counseling/investigation. Concerning the timing of patients’ follow-up, patients are managed according to specific guidelines [12,13,14]. The study cohort comprised 9 patients without thoracic aortic dilatation and 43 patients with thoracic aortic dilatation. For 10 index cases, first-degree relatives were also available (n = 35) and underwent mutational screening for variants identified in index cases (P1, P2, P4, P6, P7, P8, P9, P14, P25, P29). Among the enrolled families, nine had at least two affected family members (all except P1). Family structures (number of affected/unaffected relatives with the genotype) are shown in Figure 1 and Figure 2.

Demographic and clinical characteristics of the BAV patients are reported in Table 1. Probands whose family members have been included in the study are marked in Table 2. The experimental protocol was approved by the Local Ethical Committee. All patients underwent genetic counseling and signed a written informed consent for diagnosis and research purposes in accordance with the Helsinki Declaration.

### 2.2. Echocardiographic Evaluation

All echocardiographic examinations were conducted by senior cardiologists (S.N., S.C., M.P.F.), and aortic measurements were performed by S.N. BAV was diagnosed when only two cups were unequivocally identified in systole as previously described [2,15] and classified as type 1 (left and right cusps fusion) and type 2 (right and non-coronary cusp fusion). Aortic dimensions were assessed at end-diastole in the parasternal long-axis view at the sinuses of Valsalva and proximal ascending aorta by the leading edge method [15,16], and *Z*-scores were calculated according to age-adjusted nomograms [17]. Aortic or mitral regurgitation was evaluated and graded by multiple criteria combining color Doppler and continuous wave Doppler signals, and aortic valve stenosis was evaluated and graded by peak aortic valve velocity [15]. Echocardiographic assessment of the aortic valve calcification degree [no calcification, mild calcification (small isolated spots), moderate calcification (multiple larger spots) and severe calcification (extensive thickening and calcification of all cusps)] was also performed [18].

### 2.3. DNA Extraction, Custom Targeted NGS

Peripheral venous blood was collected in EDTA Vacutainer tubes and stored at −20 °C. Genomic DNA was extracted from blood samples using a FlexiGene Kit (Qiagen, Hilden, Germany) according to the manufacturer’s instructions. The custom targeted NGS panel included 97 genes relevant in BAV, syndromic and non-syndromic aortopathy or involved in vessel or valve growth and remodeling processes (Appendix A). Genomic DNA libraries were prepared according to the SureSelectQXT protocol (Agilent Technologies, Santa Clara, CA, USA). The pooled libraries were paired-end sequenced on a MiSeq instrument (Illumina Inc., San Diego, CA, USA). An analytical pipeline developed, implemented and validated for data analysis of targeted sequencing for diagnostic/research purposes was available in our laboratory.

### 2.4. Alignment and Variant Calling

Fastq file quality was checked with FASTQC. Trimmed reads were aligned to the human reference genome (Human GRCh37/hg19) using BWA-MEM. Bam file quality was evaluated with Qualimap 2.2.2d. Variant calling was performed using GATK4 HaplotypeCaller in GVCF mode and the joint genotyping tool GenotypeGVCFs. Variants were annotated using VEP 99. Ninety-nine percent of targeted regions were covered. Variants were filtered according to Phred quality score (Q) ≥ 30 and minimum coverage depth of 30×. Variants were called following guidelines suggested by the Broad Institute, commonly accepted as standard and identified according to (a) MAF < 0.01; (b) potential role, according to variant classification recommendation [19], literature genotype–phenotype association data and biological plausibility; (c) in silico predictor tools; (d) type of genetic variants; (e) localization (exonic, splicing region variants); and (f) allele balance > 0.2. NGS experiments showed at least 99.8% coverage and 250–300× average coverage depth; intron regions were analyzed and covered for ±50 bp from exon–intron boundaries. In the rare cases in which low coverage depth was observed in exons or flanking regions (<0.01% of total target bases on average), Sanger technology was used to further evaluate these regions.

### 2.5. Statistical Analysis

Statistical analysis was performed by SPSS package v29 (SPSS Inc.; Chicago, IL, USA). Categorical variables are expressed as frequencies and percentages, whereas continuous data are given as median and IQR. The χ^2^ test was used to compare dichotomous data. Statistical significance was accepted at *p*-value < 0.05.

## 3. Results

The global demographic and clinical characteristics and principal phenotypic features of the 52 index BAV cases are reported in Table 1 and Table 2.

As missense, in-frame indel, frameshift, nonsense and splice site/synonymous variants suggestive of alteration of the splicing process and classified as uncertain significance variants were considered, globally in the whole BAV cohort (n = 52), 194 rare variants in 63 genes were identified. In particular, among the 194 rare variants identified, 177 (91.2%) were missense (60 classified as benign, 21 as likely benign, 92 as uncertain significance, 3 as likely pathogenic, 1 as pathogenic), 3 (1.5%) in-frame indel (likely benign/benign), 3 (1.5%) frameshift (2 classified as benign and 1 as pathogenic), 2 (1%) nonsense variants (likely pathogenic), 5 (2.6%) splice site and 4 (2.1%) synonymous variants (Table 3).

Among the genes previously described to be associated with BAV and present in our NGS gene panel, five variants in the *NOTCH1* gene in four BAV patients (7.7%), four variants in the *FBN1* gene in two patients (3.8%), one variant in the *GATA5* gene (1.9%) and one variant in the *SMAD4* gene (1.9%) were identified. Furthermore, a comprehensive evaluation of the most represented variants in the BAV cohort shows a greater burden of genetic variants in the *PDIA2* gene [13 variants in 11 patients (21.1%)], *LRP1* [7 variants in 7 patients (13.5%)], *LTBP1* [10 variants in 9 patients (17.3%)], *COL6A3* [9 variants in 8 patients (15.4%)] and *LTBP2* [6 variants in 6 patients (11.5%)]; moreover, 6 patients (11.5%) carried a rare variant in *CAPN2* and *FN1*, respectively, and 5 out of the 52 BAV patients (9.6%) carried a rare variant in the *MYLK* gene.

For 10 out of the 52 BAV patients (Table 3), relatives to perform segregation analyses of the variants identified in the probands were available. In particular, 9 out of the 10 probands have at least two available affected members (all except P1).

The possible contribution of *PDIA2*, *LRP1* and *CAPN2*, suggested by the high prevalence in our patients of genetic variants with a potential effect on the phenotype of these loci in the whole cohort of BAV patients, was further explored through segregation analyses of identified variants in these genes in family members of P6, P25, P7 and P9 probands. In fact, for (1) *PDIA2* p.Pro131Leu in the P6 family and p.Thr115Met in the P25 family, (2) *LRP1* p.Val176Asp in the P7 family, (3) *CAPN2* c.901+8C>G in the P9 family, segregation data are compatible with gene involvement in the clinical phenotype, even if they do not prove a causal role (Figure 2).

Moreover, data from further segregation analyses are compatible with the role of the following variants: *MYH11* p.Ser1691Arg variant in the P8 family (Figure 1), *TGFBR3* p.Asp426Asn, *HOXA1* p.Asn269Ile and *B3GAT3* p.Ala56Val in the P14 family (Figure 1) and *SLC2A10* p.Ala385Gly in the P25 family (Figure 2).

Concerning the four likely pathogenetic and the two pathogenetic variants, they were identified in *FBN1*, *MYH11*, *MYLK*, *COL6A3*, *ABCC6* and *CBS* genes in index cases without family member availability or informative families (Table 3).

In the BAV cohort, 43 patients exhibited thoracic aortic dilatation, whereas 9 did not show the presence of aortic root as well as ascending aorta dilatation (Table 1 and Table 2). Rare variant prevalence evaluation of the targeted genes according to the presence or absence of aortic dilatation showed a statistically significant higher prevalence of *MYLK* variants in patients without aortic dilatation than in those with dilatation (33.3% vs. 4.6%; *p* = 0.03).

Fifteen patients showed the presence of aortic calcification (Table 2). Among patients with calcification, a higher prevalence of *CAPN2* rare variant carriers is observed if compared with those without calcification (26.7% vs. 5.4%, respectively; *p* = 0.05).

Among patients with stenosis, a significantly higher prevalence of *VHL* and *AGGF1* rare variant carriers was observed with respect to those without stenosis (40% vs. 0%, *p* = 0.008 and 40% vs. 0%, *p* = 0.008, respectively).

No significant difference in rare variant prevalence in the targeted genes according to the presence of absence of aortic insufficiency was found.

Rare variant distribution according to the presence or absence of thoracic aorta dilatation, calcification, aortic insufficiency and stenosis is reported in Appendix A.

## 4. Discussion

In this paper, 52 consecutive BAV patients enrolled in an Italian Referral Regional (Tuscany) Center for Marfan syndrome, Heritable Thoracic Aortic Disease (HTAD) and related disorders were studied. Mutation analysis of 97 genes regarded as relevant in BAV, syndromic and non-syndromic aortopathy or involved in vessel or valve growth and remodeling processes confirms the involvement in BAV disease of *NOTCH1*, *FBN1*, *GATA5* and *SMAD4* genes and suggests the contribution of novel genes. In particular, according to the evidence of high-prevalence rare variants with a potential effect on the phenotype in the whole cohort of the 52 BAV patients and the further segregation analyses in the 10 BAV families, the study suggested that n = 3 genes emerge as promising candidates for association in BAV, namely *PDIA2*, *LRP1* and *CAPN2*. Moreover, present data also suggest the possible contribution of rare variants in BAV complications and in particular *MYLK* in aortic dilatation, *CAPN2* in BAV calcification and *VHL* and *AGGF1* in valve stenosis.

The contribution of *PDIA2* and *LRP1* was first suggested by their high prevalence in BAV patients (21.1% *PDIA2* and 13.5% *LRP1* variant carriers) and investigated and supported by segregation analyses in families P6 and P25 (*PDIA2*) and P7 (*LRP1*), which evidence that these loci may be no longer excluded for association with the BAV phenotype. Moreover, in a non-negligible percentage of BAV patients (11.5%), rare variants in *CAPN2* were identified. In the P9 familial case, the *CAPN2* variant was also shown to segregate with the BAV phenotype, thus supporting its involvement in this phenotypic trait.

As concerns *PDIA2*, previous Genome-Wide Association Study data suggested its possible implication in BAV. In a cohort of 68 BAV probands and 830 control subjects, a haplotype within the *AXIN1-PDIA2* locus was found to be associated with BAV [21]. Moreover, data from Dargis et al. [22] reporting targeted NGS sequencing results from 48 BAV patients showed the presence of rare and common genetic variants in the *PDIA2* gene. Our data contribute to strengthening the association of *PDIA2* with BAV pathogenesis. In fact, beside the identification of 13 *PDIA2* rare variants in 11 BAV patients, segregation analysis carried out on P6 and P25 family members did not exclude the contribution of *PDIA2* mutations (p.Pro131Leu and p.Thr115Met) in framing BAV.

Data from the present study also suggested the contribution of the *LRP1* gene, encoding a multifunctional LDL receptor gene family member, in BAV predisposition; *LRP1* is actually suggested to have a major role in smooth muscle cell (vSMC) proliferation control, as well as protection against atherosclerosis [23]. Moreover, a gene expression study revealed that the *LRP1* gene was differentially expressed in both TAV and BAV subjects with dilation [24]. Interestingly, whole-exome sequencing analysis carried out on a cohort of 20 BAV subjects showed the presence of heterozygous missense mutations in 36 genes, also including *LRP1* [25]. Moreover, data from the literature also reported *LRP1* locus contribution in abdominal aortic aneurysm susceptibility [26].

Based on the abovementioned evidence, considering the different biological pathways in which LRP1 is implicated, its possible involvement in BAV should also be considered.

Rare genetic variants in the *CAPN2* gene have also been identified in 6 out of the 52 (11.5%) BAV patients investigated; *CAPN2* encodes one of the main calpain isoforms, calcium-dependent cysteine proteases able to cleave different structural proteins and involved in different intracellular signaling pathways. Data from the literature reported their contribution to angiotensin II-induced cardiovascular remodeling [27]. Moreover, increased protease activity in both MFS and BAV aortic aneurysm was detected [28]. Data from Werner et al. [29] also reported a significant difference in the expression of calpain-I and calpastatin, specifically inhibiting calpain-I and calpain-II proteolytic activity, in patients exhibiting ascending aorta aneurysm, thus supporting calpain involvement in aortic tissue structural alteration.

Moreover, data from the present study also showed a higher prevalence of *CAPN2* rare variant carriers among BAV patients with calcification with respect to the non-calcification group. Further deepening of the *CAPN2* variant effect as well as of the pathogenetic mechanisms might allow a better definition of *CAPN2*’s role in BAV development and complications.

Among BAV patients without available family members, a non-negligible percentage of subjects also carried rare variants in *LTBP1* (17.3%), *COL6A3* (15.4%), *LTBP2* (11.5%) and *FN1* (11.5%), thereby supporting the involvement of genes encoding TGF-β pathway components (i.e., LTBP, latent transforming growth factor β binding proteins) as well as the involvement of genes encoding collagen chains.

Moreover, rare variants in the *MYLK* gene, encoding myosin light chain kinase, were identified in 5 out of the 52 BAV (9.6%) patients. Although a higher percentage of *MYLK* rare variant carriers was present in BAV patients without aortic dilatation, gene variant classification according to ACMG guidelines [19] allows only one likely pathogenic variant (c.388C>T, p.Gln130*) to be identified in a subject with aortic dilatation (P51 subject, Table 3). Conversely, *MYLK* variants detected in the patient group without aortic dilatation showed a clinical significance ranging from likely benign to uncertain. Actually, the *MYLK* locus is associated with familial thoracic aortic aneurysm, type 7 (OMIM 613780), an autosomal dominant condition. Nevertheless, the role of the *MYLK* gene in the BAV phenotype should be further investigated in larger cohorts.

Among genes previously associated with BAV, five *NOTCH1* mutations were found in 4 out of the 52 (7.7%) BAV patients (P1, P30, P37, P46, Table 3). *NOTCH1* missense mutations were previously reported in the ClinVar/literature in subjects with Adams–Oliver syndrome 5 and/or left ventricular outflow tract malformation (LVOT) including BAV [30]. Previous data from targeted NGS studies also reported a *NOTCH1* mutation prevalence among BAV-unrelated subjects ranging from 1% to 9% [22,31,32]. In the present work, a higher prevalence of *NOTCH1* variants was observed in BAV patients with aortic dilatation, although the difference with respect to the patient group without aortic dilatation did not reach statistical significance. Ma et al. [32] observed among BAV patients a higher, though not statistically significant, prevalence of rare *NOTCH1* variants in subjects with normal aortas than in those with aortopathy. These conflicting data on *NOTCH1*, due to the limited number of patients investigated in the two studies, require further confirmation in larger BAV patient cohorts.

Concerning the *FBN1* gene, we found the same two missense rare variants (p.Lys2460Arg and p.Asn542Ser) in P1 and P52 (Table 3). Subject P1 represents a 45-year-old Caucasian male with a complex BAV phenotype, including thoracic aortic root and ascending aorta dilatation and connective features inconclusive for Marfan syndrome, previously reported to be investigated through a high-throughput sequencing approach [20]. In the P1 family, these variants were inherited by P1 from his mother, clinically suggestive of a fibrillinopathy responsible for a mild Marfan phenotype (Figure 2). The current study confirms this but does not add decisive new evidence regarding these specific variants. Intriguingly, subject P1 was also a carrier of variants in other genes, including *NOTCH1* [20].

Data from the present study evidencing a higher prevalence of *VHL* and *AGGF1* variants in subjects with valve stenosis also suggest the possible contribution of these loci in influencing the occurrence of this BAV complication. Actually, data from the literature show their involvement in vascular homeostasis. *AGGF1* encodes an angiogenic factor involved in the modulation of neointimal formation and restenosis associated with vascular injury due to phenotypic VSMC switching mediated by the MEK-ERK-Elk signaling pathway [33]. As concerns *VHL*, it was previously reported that germline genetic variants in the oxygen-sensing pathway (VHL-HIF2A-PHD2-EPOR) might result in erythrocytosis, thus possibly influencing hemodynamic stress on the valve [34].

The strength of our study is the availability of families for segregation analyses and homogeneity of clinical/instrumental evaluation, whereas the limitations are the low number of subjects investigated in order to reach a strong statistical power and unapplied correction for multiple comparisons, as well as the lack of a control group. Moreover, a further study limitation lies in the impossibility for NGS technology to detect large insertions/deletions in the target regions.

In conclusion, our data deriving from NGS targeted gene analysis and from segregation analyses where possible support the contribution of further loci (*PDIA2*, *LRP1*, *CAPN2*) beyond those still known (*NOTCH1*, *FBN1*, *GATA5*, *SMAD4*) to be previously associated or suspected to be associated with BAV. Nonetheless, the effect of modulatory loci that could be able to further modulate patient clinical complications (*MYLK*, *CAPN2*, *VHL* and *AGGF1*), thus contributing to delineate the genetic profile underlying the BAV phenotype, should also be considered. Nevertheless, the findings concerning *MYLK*, *CAPN2*, *VHL* and *AGGF1* genes role are exploratory and should be interpreted with caution, given the small number of events, multiple testing and the lack of correction for these tests. Moreover, variable penetrance of the trait should be responsible for lacking information from segregation analysis. Actually, a “negative” segregation with incomplete penetrance does not necessarily exclude pathogenicity; conversely, cosegregation in a small series of relatives does not represent decisive evidence.

Our results underline the complexity of clinical and genetic diagnosis in traits considered monogenic such as BAV but characterized by variability in disease phenotypic expression (incomplete penetrance), as well as the contribution of different major and modifier genes due to low/moderate or high effect genetic variants in complication development. Further evaluation in larger cohorts representative of the general BAV population as well as functional studies might allow further confirmation of the present data.

## Figures and Tables

**Figure 1 diagnostics-16-00104-f001:**
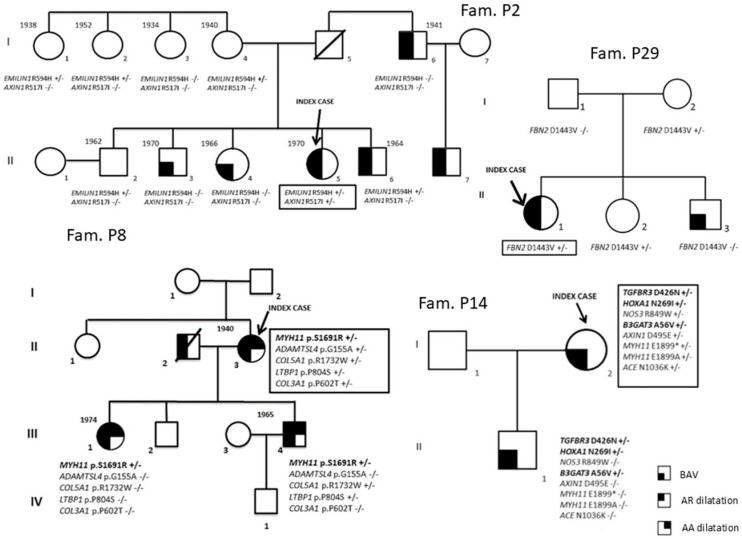
Segregation analyses (P2, P8, P14 and P29 families).

**Figure 2 diagnostics-16-00104-f002:**
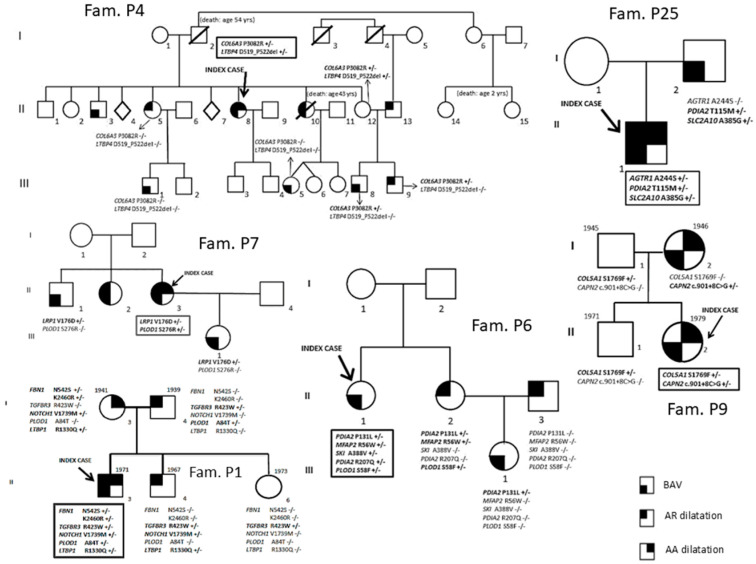
Segregation analyses (P1, P4, P6, P7, P9 and P25 families).

**Table 1 diagnostics-16-00104-t001:** Demographic and clinical characteristics of BAV patient cohort.

Feature	BAV *n* = 52
Age, years *	42.5 (31.0–51.5)
M/F	34/18
Hypertension, *n (%)*	19 (36.5)
Diabetes, *n (%)*	2 (3.8)
Dyslipidemia, *n (%)*	14 (26.9)
Smoking habit, *n (%)*	9 (17.3)
Aortic root diameter, mm *	37.0 (33.0–42.0)
Ascending aorta diameter, mm *	38.0 (34.0–45.0)
Aortic root Z-score *	1.97 (1.06–2.66)
Z-score > 2, *n (%)*	25 (48.1)
Aortic surgery, *n (%)*	1 (1.9)
BAV with root and/or ascending aorta dilatation, *n (%)*	43 (82.7)
BAV without aorta dilatation, *n (%)*	9 (17.3)
Valve types	
1 (R-L), *n (%)*	43 (82.7)
2 (R-N), *n (%)*	9 (17.3)

* *median (interquartile range, IQR)*.

**Table 2 diagnostics-16-00104-t002:** Principal phenotypic features of the 52 BAV patients.

Patient	Age at Study	Sex	Height (cm)/Weight (kg)	BAV Type	Root/AscAo (mm)	Thoracic Aorta Dilatation	Calcification	Aortic Insufficiency	Stenosis
**P1 §**	45	M	181/70	1	41/47	root/asc	Mild	Moderate	-
**P2 §**	43	F	165/57	1	36/35	root/asc	Mild	-	-
**P3**	46	F	165/61	2	33/38	asc	Mild	Mild	-
**P4 §**	52	F	165/70	1	Surgery	root/asc	Mild	Moderate	-
**P5**	33	M	185/82	1	41/40	root/asc	-	Mild	-
**P6 §**	44	F	164/61	1	29/28	-	-	-	-
**P7 §**	62	F	160/62	1	47/48	root/asc	Mild	Moderate	-
**P8 §**	70	F	160/62	1	42/48	root/asc	-	Moderate	-
**P9 §**	36	F	162/54	1	34/39	root/asc	-	-	-
**P10**	45	M	191/78	1	38/46	asc	Moderate	Mild	Mild
**P11**	54	F	153/63	1	40/53	root/asc	-	Moderate	-
**P12**	52	M	177/94	1	51/47	root/asc	Moderate	Severe	-
**P13**	13	M	149/47	1	30/26	root	Mild	Mild	-
**P14 §**	46	F	168/60	1	30/33	-	-	-	-
**P15**	26	M	178/71	1	33/38	asc	-	Mild	Moderate
**P16**	64	F	160/73	1	32/37.5	asc	Severe	Mild	Mild
**P17**	40	F	173/55	1	33/43	asc	-	-	-
**P18**	27	F	173/60	2	24/32	asc	-	Moderate	-
**P19**	46	M	168/68	2	35/34	-	Mild	Mild	-
**P20**	52	M	191/90	1	42/43	asc	Mild	Mild	-
**P21**	56	M	173/75	1	42/47	root/asc	-	Moderate	-
**P22**	25	M	183/70	1	37/45	root/asc	-	Moderate	-
**P23**	42	M	179/80	2	40/40	asc	Mild	Moderate	-
**P24**	49	M	174/78	1	48/44	root/asc	Mild	Mild	-
**P25 §**	33	M	181/78	1	43/42	root/asc	-	-	-
**P26**	41	M	173/75	2	36/44	asc	-	-	-
**P27**	31	M	180/73	1	40/44	root/asc	-	Moderate	-
**P28**	31	M	187/74	1	44/46	root/asc	-	Moderate	-
**P29 §**	17	F	164/65	1	34/34	root/asc	-	-	-
**P30**	35	M	185/90	1	46/38	root/asc	-	-	-
**P31**	22	M	190/75	1	34/34	asc	-	-	-
**P32**	44	M	184/99	1	45/47	root/asc	-	Mild	-
**P33**	50	M	181/78	1	40/40	asc	-	Mild	-
**P34**	46	F	174/63	2	39/49	root/asc	-	Mild	-
**P35**	27	M	174/64	1	39/37	root/asc	-	Mild	-
**P36**	24	M	171/66	1	37/28	root	-	Moderate	-
**P37**	33	M	182/81	2	38/38	asc	-	Mild	-
**P38**	42	M	179/74	1	44/34	root	-	-	-
**P39**	29	M	181/69	1	36/31	-	-	Mild	-
**P40**	31	M	186/75	1	36/34	-	-	Mild	-
**P41**	18	F	154/38	1	24/26	-	-	-	-
**P42**	34	F	163/66	1	29/28	-	-	-	-
**P43**	57	F	156/62	1	31/35	asc	-	-	-
**P44**	52	F	162/74	1	32/47	asc	-	-	-
**P45**	31	M	185/90	1	34/41	asc	-	Mild	-
**P46**	66	M	171/87	1	42/41	asc	Severe	-	Severe
**P47**	44	M	180/84	1	35/33	-	-	Mild	-
**P48**	50	M	171/73	1	42/37	root	-	Mild	-
**P49**	63	M	151/60	2	36/49	asc	Severe	-	Severe
**P50**	53	M	169/62	1	39/38	root/asc	-	-	-
**P51**	14	M	159/41	2	31/26	root	-	Mild	-
**P52**	35	M	175/70	1	37/32	-	-	Mild	-

- = negative; § = index cases with available family members.

**Table 3 diagnostics-16-00104-t003:** Rare genetic variants identified in the 52 BAV patients.

Patient	Gene *	Mutation cDNA	Mutation Protein	SNP Id	MAF (Eur)	SIFT	Provean	Polyphen	FATHMM	Mutation Taster	Human Splicing Finder	Splice View	Summary ^
**P1 §**	** *FBN1* **	c.7379A>G	p.Lys2460Arg	rs144189837	0.00012	T	N	PD	D	DC	-	-	LP
	** *FBN1* **	c.1625A>G	p.Asn542Ser	rs963564435	0.00001	T	N	B	D	DC	-	-	US
	** *TGFBR3* **	c.1267C>T	p.Arg423Trp	rs766001542	0.00004	T	N	B	T	P	-	-	US
	** *NOTCH1* **	c.5215G>A	p.Val1739Met	rs377294245	0.00012	D	N	PD	D	DC	-	-	US
	** *LTBP1* **	c.3989G>A	p.Arg1330Gln	rs141080282	0.00593	D	N	PD	T	DC	-	-	B
**P2 §**	** *EMILIN1* **	c.1781G>A	p.Arg594His	rs200339477	0.00044	D	N	PD	T	DC	-	-	US
	** *AXIN1* **	c.1549G>A	p.Val517Ile	rs149849071	0.00231	T	N	B	T	P	-	-	LB
**P3**													
**P4 §**	** *LTBP4* **	c.1556_1567del	p.Asp519_Pro522del	rs529188563	0.00003								B
	** *COL6A3* **	c.9245C>G	p.Pro3082Arg	rs182976977	0.00116	D	N	PD	T	P	-	-	LB
**P5**	** *FGF8* **	c.77C>T	p.Pro26Leu	rs137852660	0.00116	T	N	B	T	DC	-	-	LB
	** *AXIN1* **	c.1485C>G	p.Asp495Glu	rs146947903	0.00948	T	N	B	T	P	-	-	B
**P6 §**	** *MFAP2* **	c.166C>T	p.Arg56Trp	rs759153602	0.00004	D	N	PD	D	P	-	-	US
	** *PDIA2* **	c.392C>T	p.Pro131Leu	rs774230086	0.00012	D	D	PD	T	DC	-	-	US
	** *PDIA2* **	c.620G>A	p.Arg207Gln	rs148442250	0.00087	T	N	B	T	P	-	-	LB
	** *SKI* **	c.1163C>T	p.Ala388Val	rs75280988	0.00610	T	N	B	D	P	-	-	B
	** *PLOD1* **	c.173C>T	p.Ser58Phe	rs534978828	0.00242	D	T	PD	T	P	-	-	LB
**P7 §**	** *LRP1* **	c.527T>A	p.Val176Asp	-	N/A	D	D	PD	D	DC	-	-	US
	** *PLOD1* **	c.826A>C	p.Ser276Arg	rs760869815	N/A	T	N	B	T	P			US
**P8 §**	** *ADAMTSL4* **	c.464G>C	p.Gly155Ala	-	N/A	D	D	PD	D	DC	-	-	US
	** *MYH11* **	c.5073G>C	p.Ser1691Arg	rs760908992	0.00003	D	D	B	T	DC	-	-	US
	** *COL5A1* **	c.5194C>T	p.Arg1732Trp	rs201379514	0.00004	D	D	PD	T	DC	-	-	US
	** *LTBP1* **	c.2410C>T	p.Pro804Ser	rs149319598	0.00043	D	N	B	T	DC	-	-	US
	** *COL3A1* **	c.1804C>A	p.Pro602Thr	rs35795890	0.00656	T	D	PD	D	DC	-	-	B
**P9 §**	** *COL5A1* **	c.5306C>T	p.Ser1769Phe	rs894882071	0.00002	D	D	PD	T	DC	-	-	US
	** *CAPN2* **	901+8C>G		rs193257695	0.00327						New Site	NEG	US
**P10**	** *COL11A1* **	c.2921C>A	p.Pro974Gln	rs78046647	0.00345	T	D	PD	D	DC	-	-	B
	** *VHL* **	c.74C>T	p.Pro25Leu	rs35460768	0.00530	D	N	B	D	P	-	-	B
	** *COL2A1* **	c.3949A>G	p.Met1317Val	rs1201540358	0.00001	D	D	PD	T	DC	-	-	US
	** *COL6A1* **	c.2042T>C	p.Ile681Thr	rs138884734	0.00048	D	D	PD	T	DC	-	-	B
	** *UFD1L* **	c.601G>A	p.Glu201Lys	rs748367529	0.00006	T	N	B	T	DC	-	-	US
**P11**	** *COL11A1* **	c.3230C>T	p.Pro1077Leu	rs373734529	0.00003	D	D	PD	D	DC	-	-	US
	** *CBS* **	c.253G>A	p.Gly85Arg	rs863223435	0.00001	D	D	PD	D	DC	-	-	P
**P12**	** *CAPN2* **	c.523G>A	p.Glu175Lys	rs1030640682	0.00001	D	D	PD	D	DC	-	-	US
**P13**	** *FN1* **	c.4486C>T	p.Arg1496Trp	rs139078629	0.00772	D	D	PD	T	DC	-	-	B
	** *AXIN1* **	c.1532C>T	p.Ser511Leu	rs140190126	0.00008	T	N	B	T	P	-	-	US
	** *PDIA2* **	c.946G>T	p.Ala316Ser	rs145351921	0.00451	T	N	B	T	P	-	-	US
**P14 §**	** *TGFBR3* **	c.1276G>A	p.Asp426Asn	-	N/A	D	N	PD	T	P	-	-	US
	** *HOXA1* **	c.806A>T	p.Asn269Ile	rs760807570	0.00002	D	D	PD	D	DC	-	-	US
	** *NOS3* **	c.2545C>T	p.Arg849Trp	rs777579370	0.00002	D	D	PD	T	DC	-	-	US
	** *B3GAT3* **	c.167C>T	p.Ala56Val	rs147526003	0.00020	T	N	B	T	DC	-	-	US
	** *AXIN1* **	c.1485C>G	p.Asp495Glu	rs146947903	0.00948	T	N	B	T	P	-	-	B
	** *MYH11°* **	c.5695G>T	p.Glu1899 *	rs753304573	0.00000	-	-	-	-	-	-	-	LP
	** *MYH11°* **	c.5696A>C	p.Glu1899Ala	rs781754239	0.00000	D	D	PD	T	DC	-	-	US
	** *ACE* **	c.3108C>A	p.Asn1036Lys	rs142947404	0.00096	T	N	B	T	DC	-	-	US
**P15**	** *VHL* **	c.28G>C	p.Glu10Gln	-	N/A	D	N	PD	D	P	-	-	US
	** *AGGF1* **	c.990C>G	p.Pro330=	-	N/A	-	-	-	-	-	New Site	NEG	US
**P16**	** *TGFBR3* **	c.55A>G	p.Thr19Ala	rs147586574	0.00306	D	N	B	T	P	-	-	B
	** *TGFB2* **	c.703G>C	p.Val235Leu	rs10482810	0.00500	D	N	PD	T	DC	-	-	B
	** *MMADHC* **	c.646C>G	p.Arg216Gly	rs141093638	0.00264	T	D	B	D	DC	-	-	B
	** *COL6A3* **	c.5833G>C	p.Val1945Leu	rs113332380	0.00050	D	N	PD	T	DC	-	-	LB
	** *LTBP4* **	c.2861C>T	p.Pro954Leu	rs780185380	N/A	D	D	PD	D	D	-	-	US
	** *EMILIN3* **	c.1161A>G	p.Gln387=	rs143512004	0.00147	-	-	-	-	-	New Site	NEG	US
	** *COL6A2* **	c.272C>T	p.Ala91Val	rs201842315	0.00002	T	N	B	D	P	-	-	US
**P17**	** *COL5A1* **	c.739G>A	p.Ala247Thr	rs769115550	0.00003	T	N	PD	D	DC	-	-	US
	** *LTBP4* **	c.3098C>T	p.Pro1033Leu	-	N/A	T	N	PD	T	P	-	-	US
	** *MTR* **	c.1982T>C	p. Ile661Thr	rs148897041	0.00025	T	N	PD	T	P	-	-	LB
**P18**	** *RET* **	c.43_44insTGCTGCTGC	p.Leu19_Pro20ins LeuLeuLeu	rs768132465	0.00000	-	-	-	-	-	-	-	LB
	** *COL1A1* **	c.2005G>A	p.Ala669Thr	rs563598815	0.00001	T	N	PD	D	DC	-	-	US
	** *MAPK1* **	c.22_23insCGG	p. Ala7dup	rs751880548	0.00194	-	-	-	-	-	-	-	B
	** *LTBP1* **	c.436G>A	p.Glu146Lys	rs147166401	0.00628	T	N	B	T	P	-	-	B
	** *EMILIN1* **	c.355A>G	p.Arg119Gly	rs1572845202	N/A	D	D	PD	T	DC	-	-	US
**P19**	** *SKI* **	c.1998+6C>T	-	rs1265118842	N/A	-	-	-	-	-	New Site	New Site	US
	** *LRP1* **	c.7363G>A	p.Gly2546Ser	rs113379328	0.00376	D	D	PD	DC	D	-	-	B
	** *PDIA2* **	c.344C>T	p.Thr115Met	rs182349041	0.00532	D	N	PD	T	N	-	-	B
	** *B3GALT6* **	c.583G>C	p.Gly195Arg	rs551984021	0.00004	T	N	B	T	DC	-	-	LB
	** *TNXB* **	c.9034T>C	p.Cys3012Arg	rs2857007	0.00019	T	N	B	T	P			B
**P20**	** *CAPN2* **	c.1427A>G	p.Lys476Arg	rs9804140	0.00056	T	N	B	D	P	-	-	B
	** *LTBP1* **	c.3989G>A	p.Arg1330Gln	rs141080282	0.00593	D	N	PD	T	DC	-	-	B
	** *ZPLD1* **	c.1201A>G	p.Ser401Gly	rs144706422	0.00139	D	N	B	T	DC	-	-	US
	** *LRP1* **	c.6874C>T	p.Arg2292Cys	rs370171921	0.00002	D	D	PD	D	DC	-	-	US
	** *COL6A2* **	c.2351G>A	p.Arg784His	rs75120695	0.00712	T	N	B	T	DC	-	-	B
**P21**	** *ADAMTS2* **	c.2719G>A	p.Ala907Thr	rs199617528	0.00022	T	N	B	T	P	-	-	US
	** *LTBP3* **	c.2380G>A	p.Val794Met	rs144309426	N/A	T	N	PD	P	D	-	-	US
	** *COL2A1* **	c.1634A>G	p.Asn545Ser	rs145042175	0.00029	T	N	PD	D	DC	-	-	US
	** *COL2A1* **	c.4157C>A	p.Ser1386Tyr	-	N/A	D	D	PD	T	DC	-	-	US
	** *COL1A1* **	c.1018G>A	p.Ala340Thr	rs773343407	0.00003	T	N	B	DC	D			US
	** *PDIA2* **	c.620G>A	p.Arg207Gln	rs148442250	0.00138	T	N	B	T	P	-	-	LB
**P22**	** *COL6A3* **	c.9245C>G	p.Pro3082Arg	rs182976977	0.00115	T	N	PD	T	P	-	-	LB
	** *ADAMTS2* **	c.724G>A	p.Ala242Thr	rs372103269	0.00026	T	N	B	T	P	-	-	US
	** *LTBP2* **	c.800C>T	p.Ser267Leu	rs149952751	0.00161	T	N	B	T	P	-	-	LB
	** *COL1A1* **	c.740C>T	p.Pro247Leu	rs199626372	0.00017	D	D	B	D	DC	-	-	US
**P23 #**	** *TNXB* **	c.3322G>A	p.Val1108Met	rs121912575	0.00056	D	N	PD	T	DC	-	-	LB
	** *LTBP2* **	c.2966C>G	p.Pro989Arg	rs76172717	0.00198	D	D	PD	D	DC	-	-	B
	** *PDIA2* **	c.1055G>T	p.Gly352Val	rs45503792	0.00925	T	N	B	T	P	-	-	B
	** *PDIA2* **	c.1476_1477insA	p.Asn493Lysfs	rs45571033	0.00869	-	-	-	-	-	-	-	B
	** *CAPN2* **	901+8C>G		rs193257695	0.00327						New Site	NEG	US
**P24**	** *FN1* **	c.6047C>T	p.Pro2016Leu	rs139452116	0.00059	D	D	PD	T	DC	-	-	US
	** *COL6A3* **	c.6799G>T	p.Gly2267Cys	rs749855513	0.00000	D	D	PD	D	DC	-	-	LP
**P25 §**	** *AGTR1* **	c.730G>T	p.Ala244Ser	rs12721225	0.00388	D	N	PD	T	DC	-	-	B
	** *PDIA2* **	c.344C>T	p.Thr115Met	rs182349041	0.00532	D	N	PD	T	P	-	-	B
	** *SLC2A10* **	c.1154C>G	p.Ala385Gly	rs79849424	0.00124	T	N	B	D	P	-	-	B
**P26**	** *COL3A1* **	c.2548C>A	p.Pro850Thr	rs747312589	0.00002	D	D	PD	D	DC	-	-	US
	** *COL5A2* **	c.2237C>T	p.Pro746Leu	rs770068275	0.00002	D	D	PD	D	DC	-	-	US
	** *COL1A2* **	c.3313G>A	p.Gly1105Ser	rs139851311	0.00075	T	D	PD	D	DC	-	-	US
	** *ENG* **	c.388C>T	p.Pro130Ser	rs199840979	0.00029	T	N	B	T	P	-	-	B
**P27**	** *LTBP1* **	c.275G>A	p.Gly92Asp	rs985929702	0.00040	T	N	B	T	P	-	-	US
	** *NOS3* **	c.2479G>A	p.Val827Met	rs3918232	0.00455	D	N	PD	T	P	-	-	LB
	** *SMAD4* **	c.424+5G>A	-	rs200772603	0.00036	-	-	-	-	-	Site broken	NEG	US
**P28**	** *COL6A3* **	c.7258C>T	p.Arg2420Trp	rs150165484	0.00103	D	D	PD	D	DC	-	-	US
	** *TNXB* **	c.8132T>C	p.Ile2711Thr	rs28361051	0.0099	T	N	B	D	DC	-	-	B
	** *COL11A2* **	c.1357C>T	p.Arg453Trp	rs145499142	0.00000	D	D	PD	T	DC	-	-	B
	** *LRP1* **	c.2323G>C	p.Ala775Pro	rs34714459	0.00253	T	N	B	D	P	-	-	B
	** *PDIA2* **	c.344C>T	p.Thr115Met	rs182349041	0.00532	D	N	PD	T	P	-	-	B
	** *ABCC6* **	c.1999delG	p.Ala667Glnfs	rs72664227	0.00003	-	-	-	-	-	-	-	P
**P29 §**	** *FBN2* **	c.4328A>T	p.Asp1443Val	rs751400994	0.00016	D	D	PD	D	DC	-	-	US
**P30**	** *MFAP2* **	c.391G>A	p.Val131Ile	rs145096914	0.00027	T	N	B	D	P	-	-	US
	** *COL5A1* **	c.5136G>C	p. Leu1712=	rs760185902	0.00001	-	-	-	-	-	Site broken	NEG	US
	** *NOTCH1* **	c.1981G>A	p.Gly661Ser	rs201077220	0.00068	T	N	PD	D	DC	-	-	US
	** *NOTCH1* **	c.6577A>G	p.Ser2193Gly	rs1060502236	0.00003	D	D	PD	D	DC	-	-	US
	** *LTBP3* **	c.454G>A	p.Gly152Ser	rs200142302	0.00035	T	N	PD	T	P	-	-	US
**P31**	** *CAPN2* **	c.854G>A	p.Arg285Gln	rs755706736	0.00007	D	D	PD	D	DC	-	-	US
	** *LTBP1* **	c.436G>A	p.Glu146Lys	rs147166401	0.00650	T	N	B	T	P	-	-	B
	** *ACVR1* **	c.44C>G	p.Ala15Gly	rs13406336	0.00848	D	N	B	D	DC	-	-	B
**P32**	** *LTBP3* **	c.454G>A	p.Gly152Ser	rs200142302	0.00035	T	N	PD	T	P	-	-	US
	** *LRP1* **	c.7838G>A	p.Arg2613Gln	rs150340911	0.00203	T	N	PD	D	P	-	-	B
	** *LTBP2* **	c.4769T>C	p.Val1590Ala	rs139932140	0.00921	D	D	PD	D	DC	-	-	B
	** *ADAMTS17* **	c.2768C>G	p.Ser923Cys	rs200834418	0.00004	D	N	-	T	DC	-	-	US
	** *ACE* **	c.1174A>G	p.Met392Val	-	N/A	D	N	PD	N	DC	-	-	US
**P33**													
**P34**	** *ENG* **	c.578C>T	p.Thr193Met	rs775442178	0.00003	T	N	PD	T	P	-	-	LB
**P35**	** *COL6A3* **	c.8572G>A	p.Val2858Leu	rs111859552	0.00017	T	N	B	D	P	-	-	B
	** *COL6A3* **	c.7928C>T	p.Ala2643Val	rs111595697	0.00150	T	N	PD	D	P	-	-	B
	** *COL11A2* **	c.2078C>T	p.Pro693Leu	rs150877886	0.00100	T	D	PD	D	DC	-	-	B
	** *RET* **	c.1328A>T	p.His443Leu	-	N/A	T	N	B	T	P	-	-	US
	** *COL1A1* **	c.613C>G	p.Pro205Ala	rs72667032	0.00516	T	D	B	D	DC	-	-	B
**P36**	** *LTBP1* **	c.357C>A	p.His119Gln	rs202228164	0.00093	T	N	B	T	P	-	-	US
	** *LTBP1* **	c.3989G>A	p.Arg1330Gln	rs141080282	0.00593	D	N	PD	T	DC	-	-	B
	** *ACVR1* **	c.44C>G	p.Ala15Gly	rs13406336	0.00848	D	N	B	D	DC	-	-	B
	** *COL5A2* **	c.3689C>G	p.Thr1230Arg	rs62184175	0.00838	T	N	PD	D	DC	-	-	B
	** *PDIA2* **	c.1418G>A	p.Arg473Gln	rs116969376	0.00823	T	N	PD	T	P	-	-	LB
	** *COL6A2* **	c.943G>A	p.Asp315Asn	rs759838639	0.00002	T	N	PD	D	P	-	-	US
**P37**	** *NOTCH1* **	c.5492T>C	p.Leu1831Pro	rs1085307869	N/A	D	D	PD	D	DC	-	-	US
	** *ABCC6* **	c.4105G>A	p.Glu1369Lys	rs60285147	0.00038	D	D	PD	D	DC	-	-	US
	** *ACE* **	c.1060G>A	p.Gly354Arg	rs56394458	0.00900	D	D	PD	T	DC	-	-	B
**P38**	** *FN1* **	c.686A>C	p.Asn229Thr	-	N/A	D	D	PD	T	DC	-	-	US
	** *COL6A2* **	c.2351G>A	p.Arg784His	rs75120695	0.00712	T	N	B	T	DC	-	-	B
**P39**	** *COL6A2* **	c.2351G>A	p.Arg784His	rs75120695	0.00704	T	N	B	T	DC	-	-	B
**P40**	** *LTBP1* **	c.2503C>A	p.His835Asn	rs77938757	0.00326	T	N	B	T	P	-	-	LB
	** *MYLK* **	c.4317T>A	p.Asp1439Glu	-	N/A	D	N	B	T	DC	-	-	US
	** *EMILIN3* **	c.1432G>A	p.Glu478Lys	rs373870684	0.00004	D	D	PD	T	DC	-	-	US
	** *TGFBR2* **	c.146G>A	p.Arg49Lys	rs781529108	0.00004	T	N	B	D	P	-	-	US
	** *TNXB* **	c.6811G>T	p.Val2271Leu	rs140770834	0.00312	T	N	B	T	P	-	-	US
	** *NOS3* **	c.2512+7G>A	-	rs1405216746	0.00000	-	-	-	-	-	New Site	NEG	US
**P41**	** *COL3A1* **	c.1864C>T	p.Pro622Ser	rs772638774	0.00006	T	D	B	D	DC	-	-	US
	** *ADAMTS17* **	c.3028G>A	p.Glu1010Lys	rs367651794	0.00060	T	N	PD	T	DC	-	-	US
	** *MYLK* **	c.1968G>T	p.Trp656Cys	rs138172035	0.00214	D	D	PD	D	DC	-	-	LB
	** *EMILIN3* **	c.1838T>C	p.Leu613Ser	rs11557909	0.00598	T	N	B	T	P	-	-	LB
	** *LRP1* **	c.7985G>A	p.Arg2662Gln	rs146710883	0.00012	T	N	PD	D	P	-	-	US
	** *COL6A1* **	c.2441A>G	p.Lys814Arg	rs11553518	0.00646	T	N	B	D	P	-	-	B
**P42**	** *FN1* **	c.4036G>A	p.Glu1346Lys	rs1240216665	0.00001	D	N	PD	T	DC	-	-	US
	** *COL6A3* **	c.4280C>T	p.Pro1427Leu	rs756126826	0.00002	D	D	PD	T	P	-	-	US
	** *ADAMTS2* **	c.1736C>T	p.Thr579Met	rs760943346	0.00000	D	D	PD	T	DC	-	-	US
	** *AXIN1* **	c.1532C>T	p.Ser511Leu	rs140190126	0.00013	T	N	B	T	P	-	-	US
	** *LRP5* **	c.1738G>A	p.Val580Ile	rs149524398	0.00064	T	N	B	D	DC	-	-	US
**P43**	** *SKI* **	c.1163C>T	p.Ala388Val	rs75280988	0.00610	T	N	B	D	P	-	-	B
	** *MMADHC* **	c.617A>T	p.Asn206Ile	rs138607412	0.00143	T	N	B	D	P	-	-	US
	** *MMP2* **	c.496G>A	p.Glu166Lys	rs147947052	0.00098	T	N	B	T	DC	-	-	B
**P44**	** *MYLK* **	c.1007C>T	p.Pro336Leu	rs35912339	0.00217	D	N	B	T	P	-	-	B
	** *ENG* **	c.1096G>C	p.Asp366His	rs1800956	0.00101	D	D	PD	D	P	-	-	B
**P45**	** *COL6A3* **	c.8243C>T	p.Pro2748Leu	rs115595706	0.00016	T	N	PD	D	P	-	-	US
	** *NOS3* **	c.383G>A	p.Arg128Gln	rs148359917	0.00127	T	N	B	T	P	-	-	B
	** *RET* **	c.1399G>C	p.Val467Leu	rs200334340	0.00003	D	N	PD	D	DC	-	-	US
**P46**	** *AGGF1* **	c.367G>A	p.Glu123Lys	rs138001052	0.00001	T	D	B	T	P	-	-	B
	** *NOTCH1* **	c.2542G>A	p.Glu848Lys	rs35136134	0.00343	T	N	PD	D	DC	-	-	B
	** *PDIA2* **	c.1417C>A	p.Arg473=	rs118177911	0.0049	-	-	-	-	-	Site broken (NetGene2)	Site broken (ASSP)	US
**P47**	** *ADAMTS10* **	c.356G>C	p.Arg119Pro	rs3814291	0.00001	T	N	B	T	P	-	-	US
	** *LTBP3* **	c.2222C>T	p.Ala741Val	rs148780991	0.00499	T	N	B	T	P	-	-	US
**P48**	** *COL11A1* **	c.3811G>T	p.Val1271Leu	rs150669855	0.00009	T	N	B	D	P	-	-	LB
	** *ACVR1* **	c.44C>G	p.Ala15Gly	rs13406336	0.00848	D	N	B	D	DC	-	-	B
	** *PDIA2* **	c.1391_1392delTC	p.Leu464GlnfsTer13	rs201624048	0.00629	-	-	-	-	-	-	-	B
**P49**	** *FN1* **	c.4486C>T	p.Arg1496Trp	rs139078629	0.00772	D	D	PD	T	DC	-	-	B
	** *COL11A2* **	c.2078C>T	p.Pro693Leu	rs150877886	0.00100	T	D	PD	D	DC	-	-	B
	** *LTBP3* **	c.2222C>T	p.Ala741Val	rs148780991	0.00499	T	N	B	T	P	-	-	US
	** *LRP1* **	c.6143A>G	p.Asn2048Ser	rs1164394541	0.00001	T	N	PD	D	DC	-	-	US
	** *LTBP2* **	c.4769T>C	p.Val1590Ala	rs139932140	0.00921	D	D	PD	D	DC	-	-	B
	** *PDIA2* **	c.706G>A	p.Val236Met	rs770087656	0.00005	T	N	PD	T	P	-	-	US
	** *ACE* **	c.3656T>C	p.Leu1219Pro	rs140941300	0.00054	D	D	PD	T	D	-	-	US
	** *GATA5* **	c.698T>C	p.Leu233Pro	rs116164480	0.00207	D	D	PD	D	D	-	-	US
	** *CAPN2* **	c.202A>G	p.Ser68Gly	rs2230083	0.00072	T	N	B	T	P	-	-	B
**P50**	** *AGT* **	c.127C>T	p.Leu43Phe	rs41271499	0.00061	D	N	PD	D	D	-	-	US
	** *FN1* **	c.6982G>A	p.Gly2328Ser	rs146149463	0.00002	D	D	PD	D	D	-	-	US
	** *LTBP2* **	c.1295C>T	p.Pro432Leu	rs137854861	0.00103	T	D	B	T	P	-	-	B
**P51**	** *MTHFR* **	c.1333C>T	p.Arg445Trp	rs138469955	0.00007	D	N	B	D	DC	-	-	US
	** *MYLK* **	c.388C>T	p.Gln130 *	-	N/A	-	-	-	-	-	-	-	LP
	** *MTRR* **	c.289C>T	p.Arg97Cys	rs374239028	0.00008	T	N	B	T	DC	-	-	US
	** *NOS3* **	c.2479G>A	p.Val827Met	rs3918232	0.00455	D	N	PD	T	P	-	-	LB
	** *LTBP2* **	c.4769T>C	p.Val1590Ala	rs139932140	0.00921	D	D	PD	D	DC	-	-	B
**P52**	** *MTR* **	c.3079C>T	p.Arg1027Trp	rs116836001	0.00393	D	D	PD	T	DC	-	-	B
	** *LTBP1* **	c.1652C>G	p.Ala551Gly	rs141814525	0.00002	T	N	B	D	P	-	-	US
	** *MYLK* **	c.2461C>T	p.Arg821Trp	rs150007422	0.00033	D	N	PD	T	P	-	-	US
	** *EFEMP2* **	c.259G>A	p.Val87Ile	rs149525720	0.00003	T	N	PD	T	DC	-	-	US
	** *FBN1* **	c.7379A>G	p.Lys2460Arg	rs144189837	0.00013	T	N	PD	D	DC	-	-	LP
	** *FBN1* **	c.1625A>G	p.Asn542Ser	rs963564435	0.00001	T	N	B	D	DC	-	-	US
	** *COL1A1* **	c.4181A>G	p.Asn1394Ser	rs147266928	0.00135	D	D	B	T	DC	-	-	B
	** *COL6A1* **	c.1298G>A	p.Arg433Gln	rs151158105	0.00411	T	N	B	T	DC	-	-	LB

§ = familial cases; * = previously published data [20]; ^ = according to ACMG guidelines [19]; Eur = European (ExAC or 1000G); T = tolerated; N = neutral; D = damaging; PD = probably/possibly damaging; DC = disease causing; P = polymorphism; US = uncertain significance; LB = likely benign; LP = likely pathogenic; ° = *in cis* = final effect p.E1899S; # = patients also had a common variant c.583C>T, p.Gln195*, rs45619835, MAF = 0.01154 in the PDIA2 gene.

## Data Availability

The data that support the findings of this study are available on request from the corresponding author (E.S.).

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
