# Peer review of "Bicuspid Aortic Valve: Old and Novel Gene Contribution to Disease Onset and Complications"

_diagnostics, 2025, doi:10.3390/diagnostics16010104_

Round 1

Reviewer 1 Report

Comments and Suggestions for Authors

Lack of a proper control group.

This poses a key problem: it's impossible to accurately discuss "increased burden" of PDIA2, LRP1, CAPN2, and other variants without comparison with controls sequenced using the same method and pipeline. Bigenomic databases don't address the issue of batch effects and differences in coverage/artifacts. The authors should have clearly identified this as a major limitation, and softened the wording about "identified novel genes associated with BAV" to "candidates for further testing identified."

A Mixture of VUS and (Probably) Benign Variants in Interpretation

The "Results" and "Discussion" sections regularly use the term "rare variants" in the context of a possible contribution to pathogenesis, but most are variants of uncertain significance, while some are likely benign/benign according to the ACMG. Nevertheless, they are discussed almost equally with LP/P variants. It would be more accurate to clearly distinguish between LP/P and VUS/benign in the text; in the discussion, refer to the "VUS pool" as a hypothesis, not as the actual "genetic load" of BAV.

Multiple comparisons without correction. This significantly increases the risk of random and unstable associations. This limitation is not clearly stated.

Patients were referred to a center specializing in Marfan syndromes and related disorders; they had a high proportion of aortic dilation (82.7%). In essence, this is an enriched clinically severe sample, skewed toward genetically suspicious cases. This is described in the text, but: it is not stated that the results are poorly generalizable to the "general" BAV population, especially mild and asymptomatic forms; it is not discussed that VUS with a MAF of up to 1% in the Italian population may differ in frequency from European aggregated databases.

Inclusion criteria and "non-syndromic" BAV. The term "non-syndromic BAV" is poorly described: were all criteria assessed according to Ghent, Loeys-Dietz, and vascular EDS? What specific clinical and imaging criteria were used to exclude syndromes? How often were additional examinations performed (ophthalmology, skin and skeletal features, family history of aneurysms and dissections, etc.). Recommendation: add a subparagraph "Syndromic vs. non-syndromic classification" with clear criteria and an indication of whether there were "gray areas" (such as the aforementioned P1 with the "mild Marfan phenotype").

Positive: The pipeline is very well described (FASTQC, BWA-MEM, GATK, VEP, coverage ≥99% of target regions). What's missing: a description of the panel's "blind spots": which exons/genes had low coverage, were there any regions excluded or requiring further sensing; the possibility/impossibility of detecting CNVs, large deletions/duplications (which is important for some aortopathy genes); clarification by splice region (± how many bp from the exon-intron boundary were actually reliably covered and analyzed).

A MAF < 0.01 is used (European subsample). For BAV with a frequency of 0.5–2%, this is acceptable as a primary filter, but: it is not discussed that a variant with a MAF of ~0.5–1% in the population is unlikely to be a highly effective causal factor for monogenic BAV; a more logical approach would be to use stricter thresholds (e.g., 0.001) for interpretation as possible causal factors; or to separately analyze variants with a frequency of 0.001–0.01 as possible modifiers, but record them as such. Phenotyping of complications. There is a good volume on echocardiography, but genotype-phenotype analysis lacks: a standardized quantification of calcification (e.g., a scale or semiquantitative assessment, not simply "mild/moderate/severe" in a table); more rigorous definitions and boundaries for "stenosis" and "insufficiency" in statistical analysis (it is unclear whether all grades were combined into a single "yes/no" category); and an indication of whether age and gender were considered as covariates when comparing groups by complications.

The "Results" section already contains some wording containing interpretations ("support involvement," "suggest contribution"), which is better left to the "Discussion" section. In the "Results" section, it is optimal to limit the discussion to frequencies, variant types, and statistical test results; move any phrases about "possible role" and "supporting involvement" to the discussion.

The presence of 11-13 carriers among 52 patients is already interpreted as an argument in favor of an association. Without proper controls, this is essentially a descriptive frequency characteristic, nothing more. In the "Results" section, it would be better to: retain the facts (number of variants, types, their ACMG classes); remove wording that creates the impression of an already established relationship.

Segregation analysis: wording. The authors write that "segregation did not allow to exclude the role of the gene/variant." Formally, this is true, but: such wording sounds too "positive": essentially, it means the data are incomplete/uninformative; with a small number of informative relatives and variable penetrance, such data cannot be considered serious evidence. A better approach would be to: clearly distinguish between families with a supportive (co-segregation) pattern and uninformative/inconsistent patterns; avoid the phrase "did not allow to exclude," and instead write that "segregation data are compatible with, but do not prove a causal role."

Gene associations with complications. Problematic points: very small subgroups (9 patients without dilation, 15 with calcification, presumably even fewer with stenosis); borderline p-values ​​(0.03–0.05); MYLK: higher frequency of variants in patients without dilation, while known pathogenic MYLK variants are associated with TAA—this is an obvious contradiction that requires separate discussion as a limitation (most likely, we are talking about benign or low-impact variants). Recommendation: leave these results as exploratory/hypothesis-generating, not proven associations.

Overly strong wording about "new genes." Phrases like: "identified n=3 novel genes associated with BAV, namely PDIA2, LRP1, CAPN2" seem too categorical given: lack of controls; predominance of VUS; small sample size; lack of functional validation.

Better: talk about candidate genes or "genes that emerge as promising candidates for future validation," clearly stating that there is insufficient evidence for a causal relationship.

Insufficient criticism of one's own statistical results. It should have been explicitly stated: "The associations with MYLK, CAPN2, VHL, and AGGF1 should be interpreted with caution, given the small number of events, multiple testing, and the lack of correction for these tests; therefore, these findings are exploratory." Currently, the limitations are described softly and generally, without connection to each specific conclusion.

The role of NOTCH1 and FBN1. The discussion of NOTCH1 and FBN1 is generally adequate, but: it could be a little clearer to separate what is known from the literature from the specific contribution of this study, as it often seems that some of the conclusions are based on data from this cohort, whereas this is simply a review of the literature; For FBN1 (especially P1/P52), it would be helpful to: more clearly explain that the P1 phenotype has already been described, and that the current study confirms, but does not add decisive new evidence regarding these specific variants.

The issue of incomplete penetrance. The authors mention incomplete penetrance, but: do not discuss its implications for the interpretation of negative or inconclusive segregation; do not mention that, under such conditions, family data easily lead to errors in overestimating or underestimating the contribution of the variant. It would be worth adding a paragraph explaining that with incomplete penetrance, even "negative" segregation does not necessarily exclude pathogenicity, and vice versa—cosegregation in a small series of relatives is not decisive evidence.

What should be detailed. Subsection in "Methods" on: criteria for excluding syndromic forms; family structure (number of affected/unaffected relatives with the genotype); Limitations on family size and information content. Statistics section: clearly state that the analysis is exploratory in nature; describe whether and how adjustments for multiple comparisons were made (if not, be honest). Genetic section: clearly separate variants by ACMG classes and limit discussion of the possible pathogenetic contribution primarily to LP/P.

What could be cut. The "Discussion" section contains rather lengthy sections of the literature review (LRP1, CAPN2, calpains, AGGF1, VHL), which: in places read like a mini-review of pathophysiology; are disproportionate in length to the original data (especially with a very weak genetic signal). Rational: leave key references and a brief link to the results; remove redundant mechanistic details where the original data are minimal or questionable.

Key, understated, or underappreciated limitations of the article: lack of an original control group and reliance on population-based data in the burden analysis; Small sample size and especially small subgroups for complication analyses; multiple uncorrected statistical comparisons, making associations with complications (MYLK, CAPN2, VHL, AGGF1) strictly hypothesis-generating; bias toward VUS and likely benign variants when discussing their contribution to pathogenesis; limited information content of the segregation analysis and somewhat optimistic wording ("did not allow to exclude"); the specificity of the cohort (Italians, a referral center for syndromic aortopathy), limiting generalizability.

Author Response

Answer to Reviewer #1:

We thank Reviewer for his/her interest in our study and for his/her revision that led us to improve the paper. According to Reviewer’s suggestions, we answered point by point.

Reviewer #1:

Lack of a proper control group.

This poses a key problem: it's impossible to accurately discuss "increased burden" of PDIA2, LRP1, CAPN2, and other variants without comparison with controls sequenced using the same method and pipeline. Bigenomic databases don't address the issue of batch effects and differences in coverage/artifacts. The authors should have clearly identified this as a major limitation, and softened the wording about "identified novel genes associated with BAV" to "candidates for further testing identified."                                  

According to Reviewer suggestion, we modified the text and added this issue as a study limitation (Discussion section: page 8, lines 25-26; page 9, lines 1-2; page 12, lines 8-11).

A Mixture of VUS and (Probably) Benign Variants in Interpretation

The "Results" and "Discussion" sections regularly use the term "rare variants" in the context of a possible contribution to pathogenesis, but most are variants of uncertain significance, while some are likely benign/benign according to the ACMG. Nevertheless, they are discussed almost equally with LP/P variants. It would be more accurate to clearly distinguish between LP/P and VUS/benign in the text; in the discussion, refer to the "VUS pool" as a hypothesis, not as the actual "genetic load" of BAV.

The evaluation performed in the present work represents a hypothesis generating study aimed at identifying the potential role of genetic variants in target genes independently from their ACMG classification in modulating the clinical phenotype, also with a modulatory (low/moderate) effect, alongside the identification of causative/high effect variants.

According to Reviewer observations, we modified the Abstract text, as well as Results and Discussion sections in order to better clarify this issue (Abstract: page 2, lines 14-15; Results section: page 7, lines 15-16; Discussion section: page 8, lines 25-26; page 9, lines 1-3; pag 12, line 26; pag 13, line 1).

Multiple comparisons without correction. This significantly increases the risk of random and unstable associations. This limitation is not clearly stated.

Due to the explorative nature of the work and the limited number of subjects analysed, correction analyses for multiple comparison have not been used. According to Reviewer suggestion, we added this issue as a study limitation (Discussion section:  page 12, lines 8-11 and lines 17-20).

Patients were referred to a center specializing in Marfan syndromes and related disorders; they had a high proportion of aortic dilation (82.7%). In essence, this is an enriched clinically severe sample, skewed toward genetically suspicious cases. This is described in the text, but: it is not stated that the results are poorly generalizable to the "general" BAV population, especially mild and asymptomatic forms; it is not discussed that VUS with a MAF of up to 1% in the Italian population may differ in frequency from European aggregated databases.

We are aware with the reviewer concern; indeed studied patients are consecutive BAV patients enrolled in a Referral center for Marfan syndrome and related disorders and represent the cross-section of a patient reality where they are brought to attention for clinical evaluation and differential diagnosis aimed to confirm/exclude syndromic/familial conditions of aortopathy. Nevertheless, the evaluation of this clinical setting has allowed to have a broader representation of BAV-associated clinical complications, even if in a relatively small number of patients, thus representing a proof-of-concept study for exploring the genetic bases of BAV complications. According to Reviewer suggestion we introduced this issue in the Discussion section (page 8, lines 21-23; page 13, lines 1-3).

Inclusion criteria and "non-syndromic" BAV. The term "non-syndromic BAV" is poorly described: were all criteria assessed according to Ghent, Loeys-Dietz, and vascular EDS? What specific clinical and imaging criteria were used to exclude syndromes? How often were additional examinations performed (ophthalmology, skin and skeletal features, family history of aneurysms and dissections, etc.). Recommendation: add a subparagraph "Syndromic vs. non-syndromic classification" with clear criteria and an indication of whether there were "gray areas" (such as the aforementioned P1 with the "mild Marfan phenotype").

All patients were assessed according to Ghent, Loeys-Dietz, and vascular EDS clinical criteria. A multidisciplinary clinical evaluation is routinely performed for these patients, including cardiovascular investigation/imaging, ophthalmologist investigation, physical examination and genetic counselling/investigation. Concerning timing of patients follow-up, patients are managed according to the specific guidelines (Vahanian A, et al.; ESC/EACTS Scientific Document Group. 2021 ESC/EACTS Guidelines for the management of valvular heart disease. Eur Heart J. 2022; Praz F et al.; ESC/EACTS Scientific Document Group. 2025 ESC/EACTS Guidelines for the management of valvular heart disease. Eur Heart J. 2025; https://vascern.eu/group/heritable-thoracic-aortic-diseases-2/clinical-decision-support-tools/clinical-practice-guidelines/). This issue has been added in the Methods section (page 4, lines 11-14). Accordingly, three more references have been added.

BAV patients analysed are all non-syndromic; the only subject with few (n=3) systemic manifestations among those contributing to systemic major criterium for Marfan Syndrome (≥7), besides the major criterium of thoracic aorta aneurysm, at clinical evaluation was P1 reaching diagnosis of MFS milder phenotype after the identification of a likely pathogenetic variants in FBN1 gene.

Positive: The pipeline is very well described (FASTQC, BWA-MEM, GATK, VEP, coverage ≥99% of target regions). What's missing: a description of the panel's "blind spots": which exons/genes had low coverage, were there any regions excluded or requiring further sensing; the possibility/impossibility of detecting CNVs, large deletions/duplications (which is important for some aortopathy genes); clarification by splice region (± how many bp from the exon-intron boundary were actually reliably covered and analyzed).

NGS experiments showed at least 99,8% coverage and 250-300X average coverage depth; intron regions are analysed and covered for ±50bp from exon-intron boundaries. In the rare cases in which low coverage depth is observed in exons or flanking regions (<0.01% of total target bases in mean), Sanger technology is used to further evaluate these regions.

This information has been added in the Method section (page 6, lines 7-11).

We are aware of the possibility/impossibility of detecting CNVs, large deletions/duplications with these technology. This has been added as a further study limitation in the Discussion section (page 12, lines 10-11).

A MAF < 0.01 is used (European subsample). For BAV with a frequency of 0.5–2%, this is acceptable as a primary filter, but: it is not discussed that a variant with a MAF of ~0.5–1% in the population is unlikely to be a highly effective causal factor for monogenic BAV; a more logical approach would be to use stricter thresholds (e.g., 0.001) for interpretation as possible causal factors; or to separately analyze variants with a frequency of 0.001–0.01 as possible modifiers, but record them as such.

Among the 194 rare variants identified only 36 (18.5%) exhibited a Minor Allele Frequency higher than 0.05%.

Beyond variants with a likely pathogenic/pathogenic clinical significance, which are separately mentioned in the text, the possible contribution of other variants as causative (high effect)/modulatory (low/moderate effect) factors of the phenotype has been suggested. In order to better clarify this issue, the text throughout the manuscript has been modified (i.e. Discussion section: page 12, line 26; page 13, line 1).

Phenotyping of complications. There is a good volume on echocardiography, but genotype-phenotype analysis lacks: a standardized quantification of calcification (e.g., a scale or semiquantitative assessment, not simply "mild/moderate/severe" in a table); more rigorous definitions and boundaries for "stenosis" and "insufficiency" in statistical analysis (it is unclear whether all grades were combined into a single "yes/no" category); and an indication of whether age and gender were considered as covariates when comparing groups by complications.

More details concerning evaluation of cardiovascular clinical complications in BAV patients have been provided in the Methods section (page 5, lines 9-12). The degree of calcification of the aortic valve was scored as follows by echocardiography: 1, no calcification; 2, mildly calcified (small isolated spots); 3, moderately calcified (multiple larger spots); and 4, heavily calcified (extensive thickening and calcification of all cusps) according to Rosenhek NEJM 2000. A further reference (Reference 18) has been also added.

Due to the number of subjects investigated, the relationship between genetic variants and clinical complications severity has not been furtherly stratified with respect to that reported in the text in which all grades are combined vs no single complication. Data reported are explorative and need further confirmation in larger cohorts. Accordingly, a further sentence has been added in the Discussion section (page 13, lines 1-3).

The "Results" section already contains some wording containing interpretations ("support involvement," "suggest contribution"), which is better left to the "Discussion" section. In the "Results" section, it is optimal to limit the discussion to frequencies, variant types, and statistical test results; move any phrases about "possible role" and "supporting involvement" to the discussion.

According to Reviewer suggestion, the text in the Results section has been modified.

The presence of 11-13 carriers among 52 patients is already interpreted as an argument in favor of an association. Without proper controls, this is essentially a descriptive frequency characteristic, nothing more. In the "Results" section, it would be better to: retain the facts (number of variants, types, their ACMG classes); remove wording that creates the impression of an already established relationship.

According to Reviewer suggestion, the text in the Results section has been modified. Moreover, the lack of a control group has been added as a limitation of the study (page 12, lines 8-11).

Segregation analysis: wording. The authors write that "segregation did not allow to exclude the role of the gene/variant." Formally, this is true, but: such wording sounds too "positive": essentially, it means the data are incomplete/uninformative; with a small number of informative relatives and variable penetrance, such data cannot be considered serious evidence. A better approach would be to: clearly distinguish between families with a supportive (co-segregation) pattern and uninformative/inconsistent patterns; avoid the phrase "did not allow to exclude," and instead write that "segregation data are compatible with, but do not prove a causal role."

According to Reviewer suggestion, we modified the text in the Results section (page 7, lines 20-24).

Gene associations with complications. Problematic points: very small subgroups (9 patients without dilation, 15 with calcification, presumably even fewer with stenosis); borderline p-values (0.03–0.05); MYLK: higher frequency of variants in patients without dilation, while known pathogenic MYLK variants are associated with TAA—this is an obvious contradiction that requires separate discussion as a limitation (most likely, we are talking about benign or low-impact variants). Recommendation: leave these results as exploratory/hypothesis-generating, not proven associations.

According to Reviewer observation, we modified the text in the Discussion section (page 11, lines 3-4; page 12, lines 17-20).

Overly strong wording about "new genes." Phrases like: "identified n=3 novel genes associated with BAV, namely PDIA2, LRP1, CAPN2" seem too categorical given: lack of controls; predominance of VUS; small sample size; lack of functional validation.

Better: talk about candidate genes or "genes that emerge as promising candidates for future validation," clearly stating that there is insufficient evidence for a causal relationship.

The text in the Discussion section has been modified, according to Reviewer suggestion (page 9, lines 1-3).

Insufficient criticism of one's own statistical results. It should have been explicitly stated: "The associations with MYLK, CAPN2, VHL, and AGGF1 should be interpreted with caution, given the small number of events, multiple testing, and the lack of correction for these tests; therefore, these findings are exploratory." Currently, the limitations are described softly and generally, without connection to each specific conclusion.

As suggested by the Reviewer, this issue has been explicitly stated (Discussion section: page 12, lines 17-20;  page 13, lines 1-3).

The role of NOTCH1 and FBN1. The discussion of NOTCH1 and FBN1 is generally adequate, but: it could be a little clearer to separate what is known from the literature from the specific contribution of this study, as it often seems that some of the conclusions are based on data from this cohort, whereas this is simply a review of the literature; For FBN1 (especially P1/P52), it would be helpful to: more clearly explain that the P1 phenotype has already been described, and that the current study confirms, but does not add decisive new evidence regarding these specific variants.

According to Reviewer suggestion, this issue has been also added in the text (Discussion section: page 11, lines 22-23).

The issue of incomplete penetrance. The authors mention incomplete penetrance, but: do not discuss its implications for the interpretation of negative or inconclusive segregation; do not mention that, under such conditions, family data easily lead to errors in overestimating or underestimating the contribution of the variant. It would be worth adding a paragraph explaining that with incomplete penetrance, even "negative" segregation does not necessarily exclude pathogenicity, and vice versa—cosegregation in a small series of relatives is not decisive evidence.

As suggested by the Reviewer, a paragraph better stating that even "negative" segregation does not necessarily exclude pathogenicity, and vice versa—cosegregation in a small series of relatives is not decisive evidence has been added (Discussion section:  page 12, lines 21-23).

What should be detailed. Subsection in "Methods" on: criteria for excluding syndromic forms; family structure (number of affected/unaffected relatives with the genotype); Limitations on family size and information content. Statistics section: clearly state that the analysis is exploratory in nature; describe whether and how adjustments for multiple comparisons were made (if not, be honest). Genetic section: clearly separate variants by ACMG classes and limit discussion of the possible pathogenetic contribution primarily to LP/P.

This observations have been detailed both in the Methods and in the Discussion section throughout the text. As previously stated, the evaluation performed in the present work represents a hypothesis generating study aimed at possibly suggesting the role of genes possibly modulating the clinical phenotype rather than the identification of high penetrance causative variants. This issue has been more clearly reported throughout the text (i.e. Discussion section: page 12, line 26; page 13, line 1).

What could be cut. The "Discussion" section contains rather lengthy sections of the literature review (LRP1, CAPN2, calpains, AGGF1, VHL), which: in places read like a mini-review of pathophysiology; are disproportionate in length to the original data (especially with a very weak genetic signal). Rational: leave key references and a brief link to the results; remove redundant mechanistic details where the original data are minimal or questionable.

As suggested these paragraphs have been shortened.

Key, understated, or underappreciated limitations of the article: lack of an original control group and reliance on population-based data in the burden analysis; Small sample size and especially small subgroups for complication analyses; multiple uncorrected statistical comparisons, making associations with complications (MYLK, CAPN2, VHL, AGGF1) strictly hypothesis-generating; bias toward VUS and likely benign variants when discussing their contribution to pathogenesis; limited information content of the segregation analysis and somewhat optimistic wording ("did not allow to exclude"); the specificity of the cohort (Italians, a referral center for syndromic aortopathy), limiting generalizability.

According to Reviewer observations, these issues have been addressed throughout the text (i.e. Discussion section: page 12, lines 8-11).

Reviewer 2 Report

Comments and Suggestions for Authors

Dear Authors,

Hello,

This manuscript investigates the genetic architecture of bicuspid aortic valve (BAV) and its complications using a targeted 97-gene high-throughput sequencing approach in 52 Italian patients, complemented by segregation analysis in 10 families. The study aims to identify both previously described and novel candidate genes associated with BAV susceptibility and phenotypic complications such as thoracic aortic dilatation, calcification, and stenosis. The work is timely, clinically relevant, and adds meaningful data to the growing literature on the polygenic and heterogeneous nature of BAV.

The paper is generally well structured, with a clear methodology and thorough variant annotation. The inclusion of family segregation analyses strengthens several of the associations proposed.

However, certain areas of the manuscript would benefit from clarification:

1-The manuscript proposes three “novel” BAV-associated genes (PDIA2, LRP1, CAPN2). While gene burden and segregation data provide preliminary support, the evidence remains associative rather than causative.

The authors should explore more regarding “novel candidate genes” and emphasize the exploratory nature of the findings.

2-The authors highlight several statistically significant associations (e.g., MYLK with absence of aortic dilatation; CAPN2 with calcification; VHL/AGGF1 with stenosis).

The authors should  highlight this evidence with supportive  multiple-testing correction or discuss explicitly why correction was not performed and the implications for false-positive findings.

My recommendation :Minor revision addressing the 2 points I highlighted in the text.

Author Response

Answer to Reviewer #2:

We thank Reviewer for his/her interest in our study and for his/her revision that led us to improve the paper. According to Reviewer’s suggestions, we answered point by point.

Reviewer #2:

1-The manuscript proposes three “novel” BAV-associated genes (PDIA2, LRP1, CAPN2). While gene burden and segregation data provide preliminary support, the evidence remains associative rather than causative. The authors should explore more regarding “novel candidate genes” and emphasize the exploratory nature of the findings.

According to Reviewer suggestion, the text has been modified by emphasizing the exploratory nature of the  present findings.

2-The authors highlight several statistically significant associations (e.g., MYLK with absence of aortic dilatation; CAPN2 with calcification; VHL/AGGF1 with stenosis).

The authors should  highlight this evidence with supportive  multiple-testing correction or discuss explicitly why correction was not performed and the implications for false-positive findings.

Due to the explorative nature of the work and the limited number of subjects analysed, correction analyses for multiple comparison have not been used. According to Reviewer suggestion, we added this issue as a study limitation (Discussion section:  page 12, lines 17-20).